# Preparation of Cu_3_N/MoS_2_ Heterojunction through Magnetron Sputtering and Investigation of Its Structure and Optical Performance

**DOI:** 10.3390/ma13081873

**Published:** 2020-04-16

**Authors:** Liwen Zhu, Xiu Cao, Chenyang Gong, Aihua Jiang, Yong Cheng, Jianrong Xiao

**Affiliations:** College of Science, Guilin University of Technology, Guilin 541004, China; zhuliwen1995@163.com (L.Z.); caoxiu23@163.com (X.C.); g6288170@163.com (C.G.); jah@glut.edu.cn (A.J.); chengyong@glut.edu.cn (Y.C.)

**Keywords:** Cu_3_N/MoS_2_ films, heterojunction, magnetron sputtering, photocatalysis

## Abstract

Cu_3_N/MoS_2_ heterojunction was prepared through magnetron sputtering, and its optical band gap was investigated. Results showed that the prepared Cu_3_N/MoS_2_ heterojunction had a clear surface heterojunction structure, uniform surface grains, and no evident cracks. The optical band gap (1.98 eV) of Cu_3_N/MoS_2_ heterojunction was obtained by analyzing the ultraviolet-visible transmission spectrum. The valence and conduction band offsets of Cu_3_N/MoS_2_ heterojunction were 1.42 and 0.82 eV, respectively. The Cu_3_N film and multilayer MoS_2_ formed a type-II heterojunction. After the two materials adhered to form the heterojunction, the interface electrons flowed from MoS_2_ to Cu_3_N because the latter had higher Fermi level than the former. This behavior caused the formation of additional electrons in the Cu_3_N and MoS_2_ layers and the change in optical band gap, which was conducive to the charge separation of electrons in MoS_2_ or MoS_2_ holes. The prepared Cu_3_N/MoS_2_ heterojunction has potential application in various high-performance photoelectric devices, such as photocatalysts and photodetectors.

## 1. Introduction

Environmental problems have worsened due to the rapid consumption of fossil fuels. Thus, efficient, energy-saving, and environmentally friendly methods must be developed to solve various pollution problems [1]. Among them, photocatalysis is a widely used technology in solar energy conversion and shows good potential [2,3]. Other available methods are sewage treatment [4,5] and dye degradation [6,7]. Heterojunctions formed with 2D materials and other semiconductors have recently attracted attention from researchers due to their various potential applications. Many heterojunction materials, including semiconductors/semiconductors, metals/semiconductors, molecules/semiconductors [8,9], and multiple heterojunctions [10], have been used in photocatalysis and achieved considerable success. Heterojunction photocatalysts typically have the following advantages: strong light absorption [11], efficient charge separation and transport [12], cocatalyst effect, and strong light absorption stability [13].

Among these heterojunctions, Cu_3_N/MoS_2_ shows promise as a photocatalyst because Cu_3_N is an outstanding semiconductor material that can substantially enhance the photoelectric performance of MoS_2_ or introduce new functions into such heterojunctions. As a typical transition metal sulfide, MoS_2_ has a S–Mo–S sandwich structure combined by the van der Waals force [14,15]. Given its unique structure and photoelectric properties, MoS_2_ has attracted considerable attention from scholars [16,17] and has been widely investigated for thin film transistors [18,19], photodetectors [20,21], and solid lubrication [22,23]. Owing to its adjustable band gap, MoS_2_ can be combined with many types of semiconductors to enhance light absorption from the ultraviolet to visible light regions [7,24], increase the separation and lifetime of charge carriers, and provide potential applications in visible-light catalysis [25,26]. Cu_3_N is a widely used material because of its metal-to-semiconductor properties [27,28]. Cu_3_N has been proposed for battery materials [2,29], catalyst additives [30], spin tunnel junction [31], memory [32], and electric transport materials [33] due to its wide range of optical band gap, low temperature of thermal decomposition, and excellent chemical activity.

Heterojunctions, such as MoS_2_/ZnO [13], MoS_2_/TiO_2_ [11], and TiO_2_/WO_3_ [34], remarkably improve the electrical conductivity and optical properties of materials. However, to the best of our knowledge, Cu_3_N/MoS_2_ heterojunction materials have not been investigated. In addition, the complicated and limited preparation of most heterogeneous structures usually involves many steps and thus is not conducive to large-scale operations. Therefore, a simple and effective preparation method for Cu_3_N/MoS_2_ heterojunction is needed. Magnetron sputtering has become a popular technique due to its advantages of fast deposition speed, wide target range, good sample quality, and controllable parameters.

Our previous work investigated the photocatalytic properties of Cu_3_N/MoS_2_ composite films. In contrast, in this study, we made a more detailed and accurate study of the Cu_3_N/MoS_2_ heterojunction, and explained in more depth the reasons for the change in the band gap of the Cu_3_N/MoS_2_ heterojunction. Through the corresponding calculation, we found that the Cu_3_N/MoS_2_ heterojunction has better optical performance. In this study, Cu_3_N/MoS_2_ heterojunction was fabricated through magnetron sputtering, and its crystal structure, chemical composition, surface morphology, and band gap structure were explored.

## 2. Experimental

Cu_3_N/MoS_2_ heterojunctions were prepared on single-crystal silicon (100) and quartz substrates by using RF magnetron sputtering (JGP-450a, SKY Technology Development Co., Ltd, Shenyang, China). First, the silicon and quartz wafer substrates were separately sonicated in an acetone and ethanol solution for 15 min, rinsed with deionized water, and dried for further use. Second, the processed monocrystalline silicon and quartz wafers were placed in the substrate support, the target was installed, and the sputtering chamber was closed. Third, the sputtering chamber was evacuated to 2.5 Pa with a mechanical pump and the vacuum was driven to 1 × 10^−4^ Pa by using a turbomolecular pump. Before the experiment, the target was presputtered in Ar atmosphere for 10 min and the surface of the target was cleaned to remove the oxide. Fourth, the MoS_2_ layer was deposited on the substrate using high-purity molybdenum disulfide target (99.99%, Beijing Jingmai Zhongke Material Technology Co., Ltd., Beijing, China) in Ar atmosphere at room temperature. Total gas flow rate, vacuum chamber pressure, power, and sputtering time were set to 40 sccm, 1.0 Pa, 150 w, and 2 and 3.5 min, respectively. Fifth, a Cu_3_N layer was deposited on the MoS_2_ layer with a sputtering time of 2 min using a copper target (99.99%, Beijing Jingmai Zhongke Material Technology Co., Ltd., Beijing, China). Total gas flow rate, flow ratio of N_2_ and Ar, vacuum chamber pressure, power, and sputtering time were set to 40 sccm, 3:1, 1 Pa, 150 W, and 1.5 min, respectively. Under the same experimental conditions, a pure Cu_3_N layer was deposited on the blank substrate for 3.5 min. The deposition rates of MoS_2_ and Cu_3_N layers are 12 and 13.3 nm/min, respectively. The diameter of molybdenum disulfide target and copper target are both 60 mm. The sputtering time for the heterojunction in the scanning electron microscopy (SEM, Hitachi, Tokyo, Japan) was enlarged 10 times, and the rest of the conditions remained unchanged to easily observe the heterojunction structure. 

The surface morphology of heterojunction was characterized with a field emission scanning electron microscope (S-4800, Hitachi, Tokyo, Japan). The crystal structure of heterojunction was characterized using X-ray diffractometry (XRD, X’Pert PRO, PANalytical, Holland). Elemental characterization was conducted via X-ray photoelectron spectroscopy (XPS, Escalab, Thermo Fisher Scientific, MA, USA). The optical band gap of heterojunction was investigated with an ultraviolet-visible (UV-vis) spectrometer (UV-2600/2700, Shimadzu, Kyoto, Japan).

## 3. Results and Discussion

The XRD test patterns of MoS_2_ layer, Cu_3_N layer, and Cu_3_N/MoS_2_ heterojunction are shown in Figure 1. The 2θ in the MoS_2_ (002), (100), (101), and (110) crystal planes was found at 14.55°, 32.3°, 36.5°, and 49.4°, respectively [22,35]. The absence of other impurity peaks in the spectrum indicated the high purity of the obtained MoS_2_. Moreover, the sharp diffraction peak of MoS_2_ revealed the good crystallinity of MoS_2_. The average grain size of the heterojunction was 14.9 nm, which was smaller than the grain size observed in the SEM image. This may have been because the size of the grain aggregate observed by the SEM was not the size of the single grain; therefore, the particle size value observed by SEM was often larger than the calculated value. When the Cu_3_N layer was deposited on the MoS_2_ layer to form the Cu_3_N/MoS_2_ heterojunction, the sample still showed reflection on the (100), (101), and (110) crystal planes of MoS_2_. However, the intensity of the (100), (101), and (110) crystal planes of MoS_2_ in the Cu_3_N/MoS_2_heterojunction was weaker than that in the pure MoS_2_ layer, indicating the reduced crystal quality of the MoS_2_ layer after Cu_3_N deposition. Compared with that of the pure Cu_3_N layer, the strength of the (100) and (200) crystal planes of Cu_3_N in the Cu_3_N/MoS_2_ heterojunction decreased, possibly due to the interdiffusion of atoms that occurred at the interface between MoS_2_ and Cu_3_N and led to the reduced quality of heterogeneous crystals. 

The surface and cross-sectional morphology of the Cu_3_N/MoS_2_ heterojunction are shown in Figure 2. This surface of the MoS_2_ layer was generally smooth and flat with block particles of uniform size and no evident cracks. These characteristics were beneficial to the growth of the Cu_3_N layer on the surface of the MoS_2_ layer. Figure 2b shows the thickness of the heterojunction. The MoS_2_ and Cu_3_N layers had a thicknesses of approximately 300 and 200 nm, respectively. The boundary between the layers clearly showed the heterojunction structure. The relationship between the film deposition thickness and deposition time suggested that the Cu_3_N/MoS_2_ heterojunction was approximately 50 nm-thick. 

The distribution of elements in the Cu_3_N/MoS_2_ heterojunction and the atomic percentage of each element are shown in Figure 3. EDS data revealed that the ratio of Cu to N atoms in the heterojunction was around 1. Moreover, its chemical ratio differed from the standard ratio of Cu_3_N, possibly because of the free Cu atoms in the heterojunction. These free atoms did not combine with the N atoms to form Cu_3_N. However, fewer Mo and S atoms were detected on the surface of the heterojunction. The ratio of Mo and S atoms was about 1.87. This phenomenon occurred because MoS_2_ can be found in the lower layer, and some Mo atoms in the lower layer had moved to the surface. Meanwhile, the S atoms had difficulty in reaching the surface. The atomic ratio of Cu and N in the film Cu_3_N was 1:1, which does not meet the standard atomic ratio of Cu_3_N, in which the N element was significantly more than the Cu element. This is because during the deposition process, the ratio of N_2_ to Ar was high, and a part of the free N atoms were adsorbed between the crystal grains. However, when Mo: S = 3.11:1.66 = 1:0.53 ≈ 1.87:1, there were fewer S atoms. This may have beeen because when MoS_2_ is deposited, a part of S becomes a single substance to escape, resulting in fewer S atoms. Mo atoms that were not combined into MoS_2_ combined with MoO_3_ to oxygen (as shown in Figure 4e,f). In addition, the XPS spectrum also proved that there were a large number of Cu^+^, N^−^ (as shown in Figure 4c,d), and Mo-S bonds (as shown in Figure 4e,f) in the film. It showed that the heterojunction is mainly composed of Cu_3_N and MoS_2_, and contains only a small amount of MoO_3_ and N atoms, which does not affect the formation of Cu_3_N/MoS_2_ heterojunction. In addition, the Si atoms detected via EDS originated from the monocrystalline silicon substrate.

The chemical composition of the Cu_3_N/MoS_2_ heterojunction was characterized using XPS, and the test results are shown in Figure 4. Figure 4a,b illustrates the total spectrum of the Cu_3_N/MoS_2_ heterojunction and MoS_2_ thin film, respectively. Figure 4a depicts that the binding energies of the S2p, Mo 3d, N1s, O1s, and Cu2p peaks were 162.6, 227.8, 397.4, 530.5, and 931.3 eV, respectively [15]. Figure 4b shows that the S2p and Mo 3d peaks appeared at the binding energies of 161.5 and 229.5 eV, respectively. The O1s peak appearing in the spectrogram might be due to the inevitable atmospheric pollution in the heterojunction during transfer and testing. However, a weak S2p peak appeared at 162.6 eV possibly due to the trace S atoms sputtered out during the deposition of the remaining MoS_2_ layer in the vacuum chamber, thus causing pollution in the Cu_3_N layer. The high intensity peaks with binding energies of 931.1 and 952.1 eV corresponded to the Cu2p_3/2_ and Cu2p_1/2_ orbital peaks, respectively, as shown in Figure 4c. Figure 4d illustrates the fitting of Cu2p peaks in the Cu_3_N layer with the Cu2p_3/2_ and Cu2p_1/2_ orbital peaks at the binding energies of 932.81 and 952.76 eV, respectively [27]. In Figure 4c,d, weaker peaks appeared at the binding energies of 933.9, 952.6 eV and 934.5, 954.4 eV, which belong to Cu^2+^, which is caused by the slight oxidation of Cu^+^. Figure 4e,f depicts the Mo3d spectra in the Cu_3_N/MoS_2_ heterojunction and pure MoS_2_ layers, respectively. After the XPS spectrum of Mo3d was fitted, four peaks appeared with different intensities. The two main peaks located at 232.0 and 228.5 eV could be inferred as the Mo3d_3/2_ orbital peak in the Mo–S hybrid bond structure in MoS_2_ that characterized Mo^4+^ in MoS_2_. The two weak peaks at 223.46 and 232.95 eV could be attributed to the S2s orbital and Mo–O hybrid bond structure that indicated the formation of sulfide and partial oxidation on the film surface [23]. However, the Mo–S hybrid bond of the Mo3d orbital peak of pure MoS_2_ was located at the binding energies of 228.4 and 231.5 eV. The XPS spectra were consistent with the XRD and EDS results and indicated a large number of MoS_2_ and Cu_3_N in the heterojunction. 

Figure 5 presents the energy band diagram of the Cu_3_N/MoS_2_ interface. The valence (VBO or Δ*Ev*) and conduction (CBO or Δ*Ec*) band offsets of the interface were calculated on the basis of the XPS spectra and corresponding band gap data as follows [13]:(1)$$\Delta E_{V}=\left(E_{\mathrm{Cu}2p_{3/2}}^{{\mathrm{Cu}}_{3}N/{\mathrm{MoS}}_{2}}-E_{\mathrm{Mo}3d_{5/2}}^{{\mathrm{Cu}}_{3}N/{\mathrm{MoS}}_{2}}\right)+\left(E_{\mathrm{Mo}3d_{5/2}}^{{\mathrm{MoS}}_{2}}-E_{\mathrm{VBM}}^{{\mathrm{MoS}}_{2}}\right)-\left(E_{\mathrm{Cu}2p_{3/2}}^{{\mathrm{Cu}}_{3}N}-E_{\mathrm{VBM}}^{{\mathrm{Cu}}_{3}N}\right)$$
(2)$$\Delta E_{C}=E_{g}^{{\mathrm{Cu}}_{3}N}-E_{g}^{{\mathrm{MoS}}_{2}}-\Delta E_{V}$$
where $E_{m}^{s}$ represents the energy of characteristic m in sample s, $E_{\mathrm{VBM}}^{{\mathrm{MoS}}_{2}}$; $E_{\mathrm{VBM}}^{{\mathrm{Cu}}_{3}N}$ denotes the maximum valence band values of Mo3d_5/2_ and Cu2p_3/2_ in the MoS_2_ and Cu_3_N samples, respectively; and $E_{g}^{{\mathrm{MoS}}_{2}}$ and $E_{g}^{{\mathrm{Cu}}_{3}N}$ indicate the band gap values of the MoS_2_ and Cu_3_N samples, respectively.

According to Equations (1) and (2), the VBO and CBO between the MoS_2_ and Cu_3_N samples were 1.42 and 0.82 eV, respectively. The band gap values of MoS_2_ and Cu_3_N at 2.43 and 1.83 eV, respectively, were consistent with those measured in the US–vis transmission spectrum, and the maximum valence band values (VBM) of MoS_2_ and Cu_3_N were 1.15 and 1.40 eV, respectively [13]. The energy band diagram of the Cu_3_N/MoS_2_ heterojunction interface could be deduced on the basis of the calculated value as shown in Figure 5. The arrangement of II-type staggered band of the Cu_3_N/MoS_2_ heterojunction was conducive to the charge separation of electrons in the heterojunction because the Fermi level of Cu_3_N was higher than that of MoS_2_. After the two materials adhered to form the heterojunction, the interface electrons flowing into Cu_3_N from MoS_2_ increased the number of electrons in the Cu_3_N layer and caused additional holes in the MoS_2_ layer. These conditions changed the optical band gap. The formed heterojunction can be applied to various high-performance photoelectric devices, such as photocatalysts and photodetectors.

The UV-vis transmission spectrum and diagram of band gap calculation of the Cu_3_N/MoS_2_ heterojunction are shown in Figure 6. The transmittance of the pure Cu_3_N layer in the wavelength range of 350–650 nm was clearly improved after a heterojunction was formed between Cu_3_N and MoS_2_. According to the transmission spectrum curve of the thin film, the *E_g_* of Mo-Cu based compound region can be calculated using the optical constant law (3) and Tauc Equation (4) as follows [31]:(3)α=ln(100/T)/d
(4)$${\left(\alpha hv\right)}^{2}=A\left(hv-E_{g}\right),$$
where T represents the transmittance, *d* denotes the film thickness, α indicates the absorption coefficient, hv refers to the photon energy, and a signifies the constant. The band gap *E*_g_ could be inferred from the relationship curve between ${\left(\alpha hv\right)}^{2}$ and hv by linearly fitting the straight part of the curve as shown in Figure 6b. The optical band gaps of the Cu_3_N layer, MoS_2_ layer, and Cu_3_N/MoS_2_ Mo-Cu-based compound region were 1.83, 2.43, and 1.98 eV, respectively.

## 4. Conclusions

Cu_3_N/MoS_2_ heterojunction was prepared on silicon and quartz substrates through magnetron sputtering. Microstructure analysis revealed the good crystallinity of MoS_2_ and Cu_3_N in the heterojunction, the uniform distribution of spherical grains on the surface, the thickness of the heterojunction at approximately 50 nm, and the formation of a heterojunction structure with a clear interface. This work provided a simple and effective method for preparing heterojunctions. XPS and UV correlation spectra and data revealed that the prepared Cu_3_N/MoS_2_ heterojunction had an arrangement of II-type staggered band. The VBO and CBO of the Cu_3_N/MoS_2_ interface were 1.42 and 0.82 eV, respectively. This condition facilitated the charge separation of electrons in MoS_2_. The prepared heterojunction can be applied to various high-performance photoelectric devices, such as photocatalysts and photodetectors.

## Figures and Tables

**Figure 1 materials-13-01873-f001:**
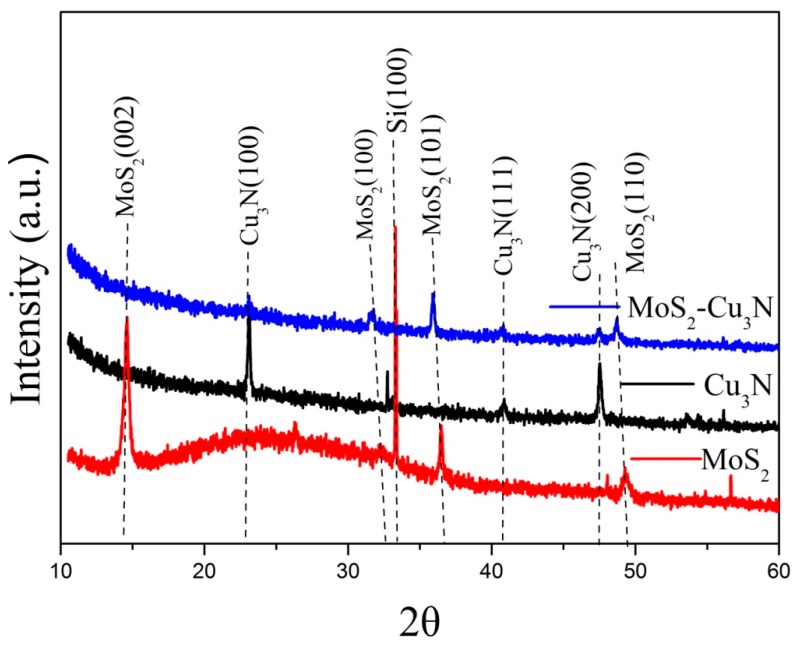
XRD patterns of the MoS_2_ layer, Cu_3_N layer, and Cu_3_N/MoS_2_ heterojunction.

**Figure 2 materials-13-01873-f002:**
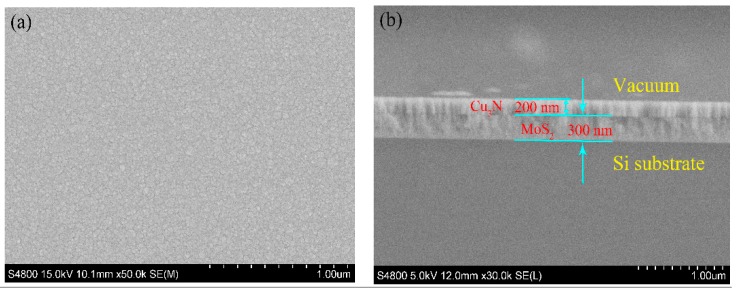
SEM image of Cu_3_N/MoS_2_ heterojunction deposited on silicon wafer: (**a**) surface of the MoS_2_ layer and (**b**) cross-section of the heterojunction.

**Figure 3 materials-13-01873-f003:**
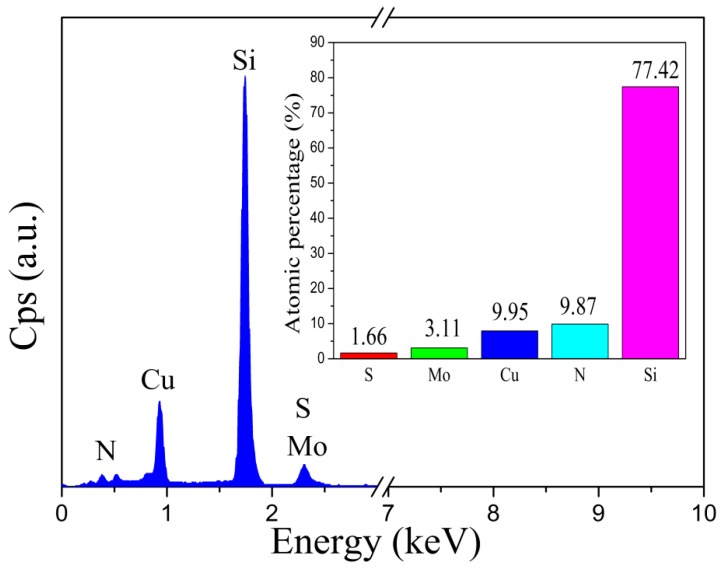
EDS spectrum of the Cu_3_N/MoS_2_ heterojunction.

**Figure 4 materials-13-01873-f004:**
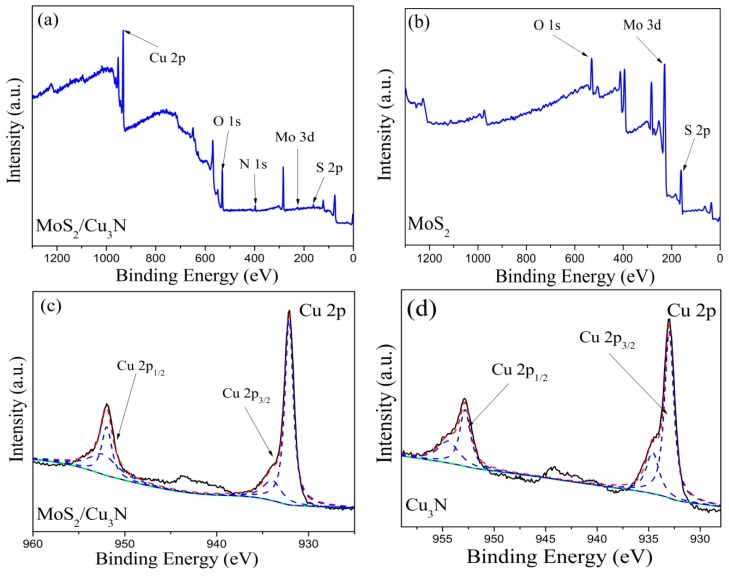
XPS spectrum: (**a**) total spectrum of the Cu_3_N/MoS_2_ heterojunction, (**b**) MoS_2_ total spectrum, (**c**) Cu 2p peak fitting in the Cu_3_N/MoS_2_ heterojunction, (**d**) Cu 2p peak fitting in the Cu_3_N layer, (**e**) Mo 3d peak fitting in the Cu_3_N/MoS_2_ heterojunction, and (**f**) Mo 3d peak fitting in the MoS_2_ layer.

**Figure 5 materials-13-01873-f005:**
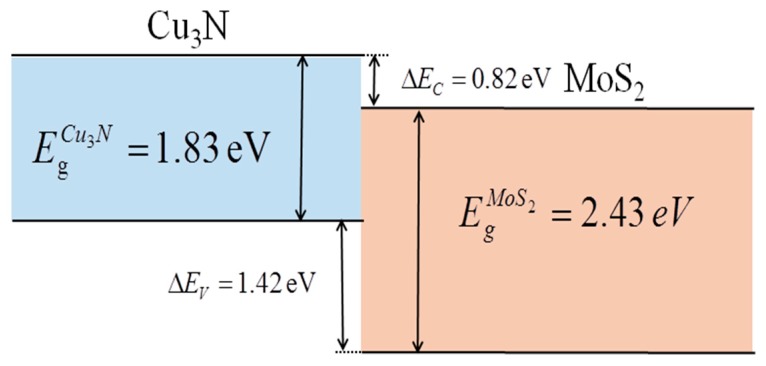
Band diagram of the Cu_3_N/MoS_2_ interface.

**Figure 6 materials-13-01873-f006:**
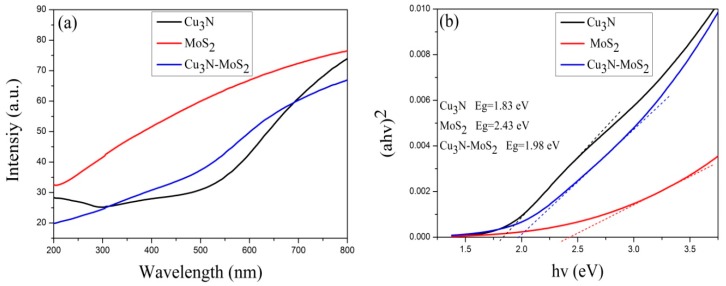
(**a**) Ultraviolet-visible (UV-vis) transmission spectrum of the Cu_3_N/MoS_2_ heterojunction and (**b**) determination of optical band gap of the Cu_3_N/MoS_2_ heterojunction.

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
