# Peer review of "Preparation of Cu3N/MoS2 Heterojunction through Magnetron Sputtering and Investigation of Its Structure and Optical Performance"

_materials, 2020, doi:10.3390/ma13081873_

Round 1

Reviewer 1 Report

The manuscript with title “Preparation of large-area Cu3N/MoS2 heterojunction through magnetron sputtering and investigation of its structure and optical band gap” discusses the fabrication of Cu3N/MoS2 heterojunction and investigates its structural and optical properties.

In the reviewer’s view, the manuscript is well-written and organized.

There are a few things that need attention to improve the overall quality of the paper.

  • The title of the manuscript may be revised to attract the readers.

  • In Experimental section, it is written that “Cu3N/MoS2 heterojunctions were prepared on single-crystal (100) and quartz substrates…” Add Silicon in the “single-crystal (100)”.
  • the MoS2 layer was deposited on the substrate using high-purity molybdenum disulfide target (99.99%) in Ar atmosphere at room temperature. Which Target was used for Cu3N layer deposition?
  • Include the deposition rates of MoS2 and Cu3N layers in the paper.
  • It is unclear the deposition of Cu3N/MoS2 heterojunction on two different substrates. Provide the reason in the paper. Was it needed for some characterization tool? What is the substrate in Fig 2b? Include it in Figure 2b and text of the manuscript. Also include scale bar in Figure 2 that is not clear.
  • High quality images for Figures 5 and 6 be provided.

As the authors have thoroughly characterized the sputter deposited Cu3N/MoS2 heterojunction with several tools and provide justification, I recommend its publication in the Materials journal after the minor corrections.

Author Response

Replies to Reviewer 1

The manuscript with title “Preparation of large-area Cu3N/MoS2 heterojunction through magnetron sputtering and investigation of its structure and optical band gap” discusses the fabrication of Cu3N/MoS2 heterojunction and investigates its structural and optical properties. In the reviewer’s view, the manuscript is well-written and organized. There are a few things that need attention to improve the overall quality of the paper.

Comment 1:The title of the manuscript may be revised to attract the readers.

Answer: Taking into account the comments of the reviewers, we revised the title of the paper: “Preparation of Cu3N/MoS2 heterojunction through magnetron sputtering and investigation of its structure and optical performance.”

Comment 2: In Experimental section, it is written that “Cu3N/MoS2 heterojunctions were prepared on single-crystal (100) and quartz substrates…” Add Silicon in the “single-crystal (100)”.

 Answer: As reviewer suggested, we have modified the paragraph in our manuscript, as follows: “Cu3N/MoS2 heterojunctions were prepared on single-crystal silicon (100) and….”.

Comment 3: The MoS2 layer was deposited on the substrate using high-purity molybdenum disulfide target (99.99%) in Ar atmosphere at room temperature. Which Target was used for Cu3N layer deposition? Include the deposition rates of MoS2 and Cu3N layers in the paper.

 Answer: The Cu3N layer is deposited with a copper target with a purity of 99.99%, and the working gases are N2 and Ar. Regarding the deposition rate of MoS2 and Cu3N layers, we have made the following additions in the article: “The deposition rates of MoS2 and Cu3N layers were 12 and 13.3 nm/min, respectively.”

Comment 4: It is unclear the deposition of Cu3N/MoS2 heterojunction on two different substrates. Provide the reason in the paper. Was it needed for some characterization tool? What is the substrate in Fig 2b? Include it in Figure 2b and text of the manuscript. Also include scale bar in Figure 2 that is not clear.

 Answer: I'm sorry, because of our negligence, did not explain in the text. We make the following instructions: We use the quartz substrate because it is transparent, which facilitates the passage of light when measuring the transmittance of the heterojunction; the single crystal silicon substrate is used because it is conductive and easy to cut, which is conducive to the testing and characterization of thin films. The substrate in Figure 2b is single crystal silicon. Regarding the scale of Figure 2, we have made corresponding changes in the paper. The picture is modified as follows:

  Fig. 2. SEM image of Cu3N/MoS2 heterojunction deposited on silicon wafer: (a) surface of the MoS2 layer and (b) cross-section of the heterojunction

Comment 5: High quality images for Figures 5 and 6 be provided.

 Answer: Thank you very much for the reminder from the reviewers, we have made changes in the paper. The picture is modified as follows:

Fig. 5. Band diagram of the Cu3N/MoS2 interface

Figure 6. (a) UV-vis transmission spectrum of the Cu3N/MoS2 heterojunction, (b) determination of optical band gap of the Cu3N/MoS2 heterojunction

Reviewer 2 Report

Submitted paper deals with the characterisation of Cu3N/MoS2 thin-film heterojunction. The films were prepared by magnetron sputtering and crystallographic structure by XRD, a composition by EDS and XPS together with UV-VIS spectroscopy were studied.

The manuscript is written in acceptable English with a proper logical structure. However, authors should comment on the novelty of this work because similar work was published recently (see doi:10.3390/coatings10010079). What is the benefit of this work? What new was discovered in comparison with previous work?

Here are my comments, which should be taken into account.

  • the title of the paper contains large-area. I did not find any relevant information in the manuscript that the authors prepared thin film on large-area substrates. They worked with small samples.
  • Experimental:
  1. I did not find magnetron diameter
  2. turbomolecular pump instead of molecular pump
  3. It has now been explained why two deposition times were used for MoS2
  4. I did not find the purity of Cu target.
  • Results and discussion:
  1. "...good crystallinity..." What does it mean? Please, be more specific, i.e. large/small crystallites, texture etc.

Author Response

Replies to Reviewer 2

Submitted paper deals with the characterisation of Cu3N/MoS2 thin-film heterojunction. The films were prepared by magnetron sputtering and crystallographic structure by XRD, a composition by EDS and XPS together with UV-VIS spectroscopy were studied.

Comment 1: The manuscript is written in acceptable English with a proper logical structure. However, authors should comment on the novelty of this work because similar work was published recently (see doi:10.3390/coatings10010079). What is the benefit of this work? What new was discovered in comparison with previous work? Here are my comments, which should be taken into account.

 Answer: Thank you very much for the reminders of the reviewers. We have made corresponding additions in the article: "In this study, we prepared Cu3N and MoS2 as a heterojunction structure and found that Cu3N/MoS2 heterojunction has better optical performance. Compared with previous work, we made a more detailed and accurate study of Cu3N/MoS2 heterojunction, explained the reasons for the change of the band gap of Cu3N/MoS2 heterojunction, and calculated the corresponding data to prove it. "

Comment 2: The title of the paper contains large-area. I did not find any relevant information in the manuscript that the authors prepared thin film on large-area substrates. They worked with small samples.

 Answer: Thanks to the reviewers for their suggestions, we have made corresponding changes in the paper.

Comment 3: Experimental:

I did not find magnetron diameter.

Answer: The target diameter is 60 mm.

Turbomolecular pump instead of molecular pump.

Answer: Thank you very much for the reviewers' comprehensive consideration of the accuracy of the article. We made the following changes in the text: “Third, …..and the vacuum was then driven to 1 × 10−4 Pa by using a turbomolecular pump.”

It has now been explained why two deposition times were used for MoS2.

Answer: The two different deposition times of MoS2 are for the purpose of comparison and subsequent experiments. The MoS2 layer with a deposition time of 2 min is part of the heterojunction, and the pure MoS2 film with a deposition time of 3 min and 30 s is the control group. When analyzing the band gap structure of the heterojunction later, we need the relevant data of pure MoS2 film for calculation and analysis. In addition, when calculating the band gap value, the comparison of the two values can more intuitively see the change of the band gap.

I did not find the purity of Cu target.

Answer: Thank you very much for the reminders of reviewers, we add the following instructions: “Fifth, a Cu3N layer was deposited ……using a copper target(99.99%, Beijing Jingmai Zhongke Material Technology Co., Ltd).”

Comment 4: Results and discussion: "...good crystallinity..." What does it mean? Please, be more specific, i.e. large/small crystallites, texture etc.

 Answer: Taking into account the comments of the reviewers, we have made corresponding additions:“The average grain size of the heterojunction was 14.9 nm, which is smaller than the grain size observed in the SEM image. This may be because the size of the grain aggregate observed by the SEM is not the size of the single grain,therefore, the particle size value observed by SEM is often larger than the calculated value.”

Reviewer 3 Report

Manuscript number: Materials-769152

Title: Preparation of large-area Cu3N/MoS2 heterojunction through magnetron sputtering and investigation of its structure and optical band gap

The authors described the obtaining of Cu3N/MoS2 heterojunction through magnetron sputtering and investigation of its structure and optical band gap.

This study is a continuation of the article entitled “Photocatalytic Properties of Copper Nitride/Molybdenum Disulfide Composite Films Prepared by Magnetron Sputtering”. It would have been fair for the authors to present these results in comparison with those already published. The synthesis procedure and the properties of material should have been compared, and the novelty must be highlighted. This is a serious problem which must be solved.

For the original part of the manuscript, after it will be highlighted, I have some observations:

  1. The authors must write “MoS2/Cu3N” or “Cu3N/MoS2" heterojunction, but not both, as it is now.
  2. In “Experimental” the reagents and their producers must be provided very clear. The experimental procedure, with references and highlighting the differences, must be described clearly and more concise.
  3. The XRD patterns must be refined and all peaks must be indexed in Fig. 1.
  4. “EDS data revealed that the ratio of Cu to N atoms in the heterojunction was around 1:3.6.” How did the authors calculate this ratio, if Cu:N ratio from the atomic percentage is almost 1 (in Fig. 3)?
  5. “minimal Mo and S atoms were detected on the heterojunction surface with more Mo atoms than S atoms”. The authors must give the atomic ratio. If the found atomic ratio did not confirm the proposed formula, which are the arguments for the "Cu3N/MoS2 heterojunction"?
  6. “The O1s peak appearing in the spectrogram might be due to the inevitable atmospheric pollution in the heterojunction during transfer and testing.” – Atmospheric pollution?
  7. Few references are required for equations.

Author Response

Replies to Reviewer 3

Title: Preparation of large-area Cu3N/MoS2 heterojunction through magnetron sputtering and investigation of its structure and optical band gap. The authors described the obtaining of Cu3N/MoS2 heterojunction through magnetron sputtering and investigation of its structure and optical band gap.

Comment 1: This study is a continuation of the article entitled “Photocatalytic Properties of Copper Nitride/Molybdenum Disulfide Composite Films Prepared by Magnetron Sputtering”. It would have been fair for the authors to present these results in comparison with those already published. The synthesis procedure and the properties of material should have been compared, and the novelty must be highlighted. This is a serious problem which must be solved.

Answer: Many thanks to the reviewers for their reminders, we have made corresponding additions in the article: "Our previous work studied the photocatalytic properties of Cu3N/MoS2 composite thin films.In contrast, in this study, we made a more detailed and accurate study on the structure of Cu3N/MoS2 heterojunction, and explained the reasons for the change of the band gap of Cu3N/MoS2 heterojunction, and calculated the corresponding data to prove it. It was found that the Cu3N/MoS2 heterojunction has better optical performance. "

Comment 2: For the original part of the manuscript, after it will be highlighted, I have some observations:The authors must write “MoS2/Cu3N” or “Cu3N/MoS2” heterojunction, but not both, as it is now.

Answer: We are very grateful for the reminders of the reviewers, we have revised all these nouns to: “Cu3N/MoS2

Comment 3: In “Experimental” the reagents and their producers must be provided very clear. The experimental procedure, with references and highlighting the differences, must be described clearly and more concise.

Answer: We are very sorry that we did not explain in the text because of our negligence. We make the following modifications: “Fourth, the MoS2 layer was …..molybdenum disulfide target (99.99%, Beijing Jingmai Zhongke Material Technology Co., Ltd) in Ar …….”. “Fifth, a Cu3N layer was deposited on the MoS2 layer with a sputtering time of 2 min using a copper target(99.99%, Beijing Jingmai Zhongke Material Technology Co., Ltd).”. “The deposition rates of MoS2 and Cu3N layers are 12 and 13.3 nm/min, respectively.”

Comment 4: The XRD patterns must be refined and all peaks must be indexed in Fig. 1.

Answer: We have revised it in the paper, the revised picture is as follows:

Fig. 1. XRD patterns of the MoS2 layer, Cu3N layer,

and Cu3N/MoS2 heterojunction

Comment 5: “EDS data revealed that the ratio of Cu to N atoms in the heterojunction was around 1:3.6.” How did the authors calculate this ratio, if Cu:N ratio from the atomic percentage is almost 1 (in Fig. 3)?

Answer: Thank you very much for the reminder of the reviewers, because our negligence caused the wrong result. We have made the following changes in the text: “EDS data revealed that the ratio of Cu to N atoms in the heterojunction was around 1.”

Comment 6: “minimal Mo and S atoms were detected on the heterojunction surface with more Mo atoms than S atoms”. The authors must give the atomic ratio. If the found atomic ratio did not confirm the proposed formula, which are the arguments for the "Cu3N/MoS2 heterojunction"?

Answer: Taking into account the recommendations of the reviewers, we made the following changes in the text: “However, fewer Mo and S atoms were detected on the surface of the heterojunction. The ratio of Mo and S atoms is about 1.87.” Regarding the argument for supporting "Cu3N/MoS2 heterojunction", in addition to EDS, we also performed XRD and XPS characterization of the heterojunction, and comprehensively analyzed the crystal structure, element composition and element chemical state of the Cu3N/MoS2 heterojunction , Thus providing an argument for " Cu3N/MoS2 heterojunction". The XRD and XPS studies are explained in detail in the paper.

Comment 7: “The O1s peak appearing in the spectrogram might be due to the inevitable atmospheric pollution in the heterojunction during transfer and testing.” – Atmospheric pollution?

Answer: We are very grateful to the reviewers for their comprehensive consideration to improve the accuracy of the paper. We make the following statement: Air pollution here refers to the pollution caused by the adsorption of oxygen in the air by the sample during the transfer and testing process.

Comment 8: Few references are required for equations.

Answer: We have added references to the equations in the paper.

Reviewer 4 Report

Paper "Prepartion of large-area..." described very interesting results of Cu and Mo based thin film junction. Unfortunetely some parts of paper are poorly prepared and need correction:

  1. Experimental section. "heterjunction were prepared on single-crystal (100)...". What type of single crystal? I mean, that silicon.
  2. What target do you use for deposition of Cu3N film? Pure Cu?
  3. In fig. 2 you presented SEM images of crossection. Have you performed EDS mapping? It will be better, than results presenten in Fig. 3.
  4. XPS results. Why you don't use reverse BE scale? From higher values down to smaller?
  5. Fig. 4c and d. XPS analysys for Cu29 region need correction. In my opinion there it should be fitted by two doublets. From with compound could be related another Cu doublet?
  6. What do you mean if you write band gap of a heterojunction? It is band gap of Mo-Cu based compond region. But not band gap of a junction.

This paper can be bublished after major revision.

Author Response

Replies to Reviewer 4

Paper "Prepartion of large-area..." described very interesting results of Cu and Mo based thin film junction. Unfortunetely some parts of paper are poorly prepared and need correction:

Comment 1: Experimental section. "heterjunction were prepared on single-crystal (100)...". What type of single crystal? I mean, that silicon.

Answer: As reviewer suggested, we have modified the paragraph in our manuscript, as follows: “Cu3N/MoS2 heterojunctions were prepared on single-crystal silicon (100) and….”

Comment 2: What target do you use for deposition of Cu3N film? Pure Cu?

Answer: Thank you very much for the reminder of the reviewers, we have made the following additions in the text: “Fifth, a Cu3N layer was deposited on the MoS2 layer with a sputtering time of 2 min using a copper target (99.99%, Beijing Jingmai Zhongke Material Technology Co., Ltd).”

Comment 3: In fig. 2 you presented SEM images of crossection. Have you performed EDS mapping? It will be better, than results presented in Fig. 3.

Answer: The reviewers have made a very good suggestion, but we are very sorry, we have not implemented EDS mapping. We cannot implement it in this article, and we will follow this recommendation in future research work.

Comment 4: XPS results. Why you don't use reverse BE scale? From higher values down to smaller?

Answer: Taking into account the comments of the reviewers, we modified the XPS spectra in the paper, and the revised results are as follows:

Fig. 4. XPS spectrum: (a) total spectrum of the Cu3N/MoS2 heterojunction, (b) MoS2 total spectrum, (c) Cu 2p peak fitting in the Cu3N/MoS2 heterojunction, (d) Cu 2p peak fitting in the Cu3N layer, (e) Mo 3d peak fitting in the Cu3N/MoS2 heterojunction, and (f) Mo 3d peak fitting in the MoS2 layer

Comment 5: Fig. 4c and d. XPS analysis for Cu2p region need correction. In my opinion there it should be fitted by two doublets. From with compound could be related another Cu doublet?

Answer: Thank you very much for the reviewer's law proposal, we have refitted Figures 4c and d, and made a supplementary explanation in the paper: "In Figures 4c and d, weaker peaks appeared at the binding energies of 933.9, 952.6 eV and 934.5, 954.4 eV, which belong to Cu2+, which is caused by the slight oxidation of Cu+.". The modified spectrum is as follows:

Fig. 4. XPS spectrum: (c) Cu 2p peak fitting in the Cu3N/ MoS2 heterojunction,

(d) Cu 2p peak fitting in the Cu3N layer

Comment 6: What do you mean if you write band gap of a heterojunction? It is band gap of Mo-Cu based compound region. But not band gap of a junction.

Answer: We are very sorry for the inaccurate statement caused by our negligence. Now, we have made the following changes in the paper: “According to the transmission spectrum curve of the thin film,…. the Eg of Mo-Cu based compond region can be calculated…. and Cu3N/MoS2 Mo-Cu based compond region were 1.83, 2.43, and 1.98 eV, respectively.”

Round 2

Reviewer 2 Report

The manuscript was substantially improved.

Author Response

Thank you very much for your comments and suggestions, and we would like to express our great appreciation to you.

Thank you and best regards.

Yours sincerely,

Jianrong Xiao

Reviewer 3 Report

Manuscript number: Materials-769152R

Title: Preparation of large-area Cu3N/MoS2 heterojunction through magnetron sputtering and investigation of its structure and optical band gap

The authors highlighted the originality of their work in comparison with their previously published article.

I have the same question regarding the formulation of heterojunction as “Cu3N/MoS2”. The proposed formula involves the atomic ratios Cu:N = 3:1 and Mo:S = 1:2. But the found atomic ratios are: Cu:N = 1:1 (in text and in Fig. 3), and  Mo:S = 1.87 (in text) respective Mo:S = 3.11:1.66 = 1:0.53 (in Fig. 3).

So, the formulation of heterojunction is not sustained by EDS results.

My question still remains:  “If the found atomic ratio did not confirm the proposed formula, which are the arguments for the formulation of heterojunction as Cu3N/MoS2?”

Author Response

The authors highlighted the originality of their work in comparison with their previously published article. I have the same question regarding the formulation of heterojunction as “Cu3N/MoS2”.

The proposed formula involves the atomic ratios Cu:N = 3:1 and Mo:S = 1:2. But the found atomic ratios are: Cu:N = 1:1 (in text and in Fig. 3), and  Mo:S = 1.87 (in text) respective Mo:S = 3.11:1.66 = 1:0.53 (in Fig. 3). So, the formulation of heterojunction is not sustained by EDS results.

My question still remains:  “If the found atomic ratio did not confirm the proposed formula, which are the arguments for the formulation of heterojunction as Cu3N/MoS2?”

Answer: Taking into account the doubts of the reviewers, we add the following explanation:

The atomic ratio of Cu and N in the film Cu3N is 1: 1, which does not meet the standard atomic ratio of Cu3N, in which the N element is significantly more than the Cu element. This is because during the deposition process, the ratio of N2 to Ar is high, and a part of the free N atoms are adsorbed between the crystal grains, which is reflected in the XPS spectrum (XPS spectrum is shown below).

However, Mo: S = 3.11: 1.66 = 1: 0.53 ≈ 1.87: 1, there are fewer S atoms. This may be because when MoS2 is deposited, a part of S becomes a single substance to escape, resulting in fewer S atoms. Mo atoms that are not combined into MoS2 combine with MoO3 to oxygen (as shown in Figure 4e and f).

In addition, the XPS spectrum also proves that there are a large number of Cu+, N- (as shown in Figure 4c and d) and Mo-S bonds (as shown in Figure 4e and f) in the film. XRD also confirmed the existence of Cu3N and MoS2, and no other impurity peaks existed. According to the area of the valence peak of each element in the XPS spectrum, the heterojunction is mainly composed of Cu3N and MoS2, and contains only a small amount of MoO3 and N atoms, which does not affect the formation of Cu3N / MoS2 heterojunction.

Reviewer 4 Report

Authors corrected manuscript according to reviewer suggestions. Paper is well prepared and I advice to publish it.

Author Response

(The authors gave the same response as above.)
